# Research on Network Attack Traffic Detection Hybrid Algorithm Based on UMAP-RF

**Xiaoyu Du [1,2], Cheng Cheng [1], Yujing Wang [1,*] and Zhijie Han [3]**

[1] School of Computer and Information Engineering, Henan University, Kaifeng 475004, China;
dxy@henu.edu.cn (X.D.); 104753200831@henu.edu.cn (C.C.)
[2] Henan Engineering Research Center of Spatial Information Processing, Kaifeng 475004, China
[3] School of Software, Henan University, Kaifeng 475004, China; hanzj@henu.edu.cn
[*] Correspondence: yjwang@henu.edu.cn; Tel.: +86-037-123-883-088

**Abstract:** Network attack traffic detection plays a crucial role in protecting network operations and services. To accurately detect malicious traffic on the internet, this paper designs a hybrid algorithm UMAP-RF for both binary and multiclassification network attack detection tasks. First, the network traffic data are dimensioned down with UMAP algorithm. The random forest algorithm is improved based on parameter optimization, and the improved random forest algorithm is used to classify the network traffic data, distinguishing normal data from abnormal data and classifying nine different types of network attacks from the abnormal data. Experimental results on the UNSW-NB15 dataset, which are significant improvements compared to traditional machine-learning methods, show that the UMAP-RF hybrid model can perform network attack traffic detection effectively, with accuracy and recall rates of 92.6% and 91%, respectively.

**Keywords:** internet; cyber attack; random forest; UMAP; machine learning

## 1. Introduction

Since 2020, with Corona Virus Disease 2019 sweeping the world, people have had to work online, and most schools have adopted online classes to teach. Internet traffic has increased dramatically, and many unscrupulous individuals have used hacking techniques to create malicious traffic to interfere with the normal operation of network devices to profit from this period. Cyber threats have been ranked as one of the most critical threats to the world economy, with expected USD 133 billion cyber-security-related expenditure to date [1]. With the constant development and popularity of cyber security [2], cyber attack security detection systems are widely used to protect cyberspace and ensure that people can browse the web in a secure network environment. Network attack traffic detection has been one of the main methods to enhance network security in recent years [3], which monitors network traffic or suspicious activities in hosts and issues alerts when such activities are detected. Currently, network attack traffic detection is gradually moving toward intelligence, and research on strange traffic attacks based on machine learning or data mining [4] has yielded significant results.

The network traffic in the current internet environment is growing, and the variety of malicious traffic is increasing. Traditional machine-learning methods are difficult to effectively detect network attack traffic and have low detection efficiency. To better handle the high-dimensional network security connection data and improve the efficiency of subsequent detection of abnormal data, we propose a hybrid algorithm—called uniform manifold approximation and projection algorithm-random forest (UMAP-RF)—for detection methods. First, large and complex network traffic is dimensionally reduced by using the uniform manifold approximation and projection algorithm (UMAP) [5], and then, an improved random forest (RF) is adopted for more accurate detection and classification.

Compared with traditional machine-learning methods, the UMAP-RF hybrid algorithm greatly reduces the detection time for network data with large and tedious data volume and improves the efficiency of network attack security detection. The characteristics of network connection data after dimensionality reduction are more pronounced, which further enhances the accuracy of differentiation based on normal data and abnormal data. This hybrid algorithm solves the problems of ineffective processing of high-dimensional data and low detection accuracy in the current research of network attack traffic detection technology, and it directly improves the accuracy and efficiency of network traffic attack detection. Moreover, this paper visualizes and analyzes the network traffic data after detection and classification, which visually verifies the effectiveness of the algorithm more intuitively. The dataset used in the experiments of this paper is UNSW-NB15 [6]. Compared with KDDCUP99 [7] and NSL-KDD [8], the dataset UNSW-NB15 has more attack types, can better simulate the network traffic on the internet nowadays and has a more practical reference value. To make the proposed solution more applicable in network attack traffic detection techniques, the KDDCUP99 and NSL-KDD datasets are used in the binary classification experimental part under the same experimental environment, and the experimental results show that the hybrid algorithm UMAP-RF still has excellent experimental results.

The main contributions of this paper are as follows. Firstly, this paper uses the UMAP algorithm based on network traffic feature filtering for dimensionality reduction, so that the reduced network security connection data highlight the key feature information, which helps classify the normal and abnormal data. Additionally, the processing speed is very fast when facing a large amount of data, which directly improves the efficiency of network security attack detection. Second, this paper restores the structure of the classified data to the maximum extent for the visualization study, and it can be observed that there is a clear distinction between normal and abnormal data in the visualization effect diagram of binary classification and multi-classification, and different kinds of abnormal data are also distinguished from each other. Third, the random forest algorithm is improved based on parameter optimization, using the number of base evaluators and the maximum number of leaf nodes to optimize the parameters of the random forest algorithm, and the improved algorithm can effectively prevent overfitting and improve the classification performance of the random forest algorithm. The accuracy and false alarm rate are significantly improved compared to other machine-learning algorithms.

Section 2 of this paper introduces the current status of domestic and international research on network attack traffic detection techniques and compares the accuracy of various detection models. Section 3 gives the hybrid algorithm UMAP-RF proposed in this paper, including the model and algorithm steps, and Section 4 conducts the experiments to evaluate and compare the accuracy and running time obtained from the detection classification.

## 2. Related Works

Machine-learning-based network attack traffic detection has attracted the attention of many researchers engaged in the network security industry [9], especially the prevalent deep learning nowadays. Several researchers have introduced deep-learning models to network attack traffic detection and achieved good results.

Ever since data mining was introduced in 1989, the application of data mining in network attack traffic detection systems has become the main research direction of network attack traffic detection technology. The U.S. Department of Defense Advanced Planning Agency (DARPA) created the DARPA 1998 dataset [10]. Subsequently, Wenke Lee et al. divided this dataset into training data with markers and unmarked test data [11], named the KDDCUP99 dataset, and later, the NSL-KDD dataset originated from it. Since the birth of these datasets, machine-learning and later deep-learning techniques have been massively applied in the study of network attack traffic detection models. Currently, the primary methods applied in machine learning for network traffic attack detection include

support vector machines (SVM) [12], decision trees [13], Bayesian [14] and artificial neural networks [15], etc. The NSLKDD dataset cannot meet the current research needs in the field of intrusion detection due to its inherent shortcomings. Moustafa et al. published the UNSW-NB15 dataset. Compared with the KDDCUP99 and NSL-KDD datasets, the UNSW-NB15 dataset contains the most comprehensive attack scenarios. Researchers can detect malicious traffic by imitating the actual network environment based on this dataset, which promotes the development of network attack traffic detection research.

More and more research on network traffic anomaly detection has been conducted at home and abroad in recent years. Guoyan Huang's team proposed a K-Means algorithm-based clustering of typical network traffic in the UNSW-NB15 dataset [16], processed the extracted representative feature subsets with a feature recursive elimination algorithm and then designed nine algorithm combinations by combining three machine-learning algorithms: decision trees, random forest and XGBoost [17]. Zhang Renjie of Nanjing University of Posts and Telecommunications clustered some samples screened by the KNN algorithm with the DBSCAN algorithm [18]. The recall and F1 scores obtained by training on some features of the UNSW-NB15 dataset showed a significant improvement. Fengjie Hu of Xidian University designed a network intrusion detection system based on the light GBM model. The accuracy obtained by comparison and validation on the dataset UNSW-NB15 was 85.78% [19]. Meftah experimented with an improved SVM algorithm on the UNSW-NB15 dataset, and the accuracy obtained was 82% [20]. Kasongo improved the test accuracy of the binary classification scheme from 88.13% to 90.85% by using the XGBoost-based feature selection method allowing DT and other ways [21]. Cao Bo proposed a network intrusion detection model incorporating a convolutional neural network and gated recursive units and obtained an accuracy of 86.25% after experiments on the UNSW-NB15 dataset, which was 1.95% higher than the same type of CNN-GRU [22]. Alzaqebah tuned the parameters of the extreme learning machine (ELM) by an improved gray wolf optimization algorithm (GWO) to test the proposed method using the UNSW-NB15 dataset and experimentally obtained an accuracy rate of 81% [23].

However, these methods did not perform a good job in dimensionality reduction in the dataset. The training speed of traditional machine-learning algorithms will be significantly reduced when facing large network traffic. For such problems, the UMAP algorithm based on network traffic feature filtering can effectively handle high-dimensional network traffic data and make the characteristics of the reduced-dimensional network traffic data more pronounced. It has a speedy running time and processes the data with high computational efficiency. Additionally, the hybrid algorithm UMAP-RF yields an accuracy of 92.6% after binary classification experiments on the UNSW-NB15 dataset, both of which are more accurate and have better performance than those obtained by the methods proposed in the above literature.

## 3. Hybrid Algorithm UMAP-RF

The hybrid algorithm UMAP-RF performs dimensionality reduction on the UNWS-NB15 dataset by the UMAP algorithm, and the reduced data are classified by the improved random forest algorithm based on parameter optimization to distinguish normal data from abnormal data and then classify the abnormal data.

### 3.1. UMAP Dimensionality Reduction Algorithm

The UMAP algorithm was created based on the theoretical framework structure of Riemannian geometry and algebraic topology [5] to reduce the dimensionality of high-dimensional data based on the conclusion that high-dimensional spaces map to low-dimensional similarities [24].

**Theorem 1 .** *In Euclidean space, mapping points in high-dimensional space to low-dimensional space, the points originally close are definitely still close in low-dimensional space, but the points originally far away have some probability of becoming close.*

**Proof of Theorem 1.** In the process of mapping points in high-dimensional space to low-dimensional space, we need to use embedding to calculate the similarity using inner product operation. Suppose $v$ is a $k$-dimensional embedding vector in high-dimensional space, and $x$ is a randomly generated $k$-dimensional mapping vector. Then, we can use the inner product operation to map $v$ to a one-dimensional space and obtain the value $h(v) = v * x$. In the process of mapping, some distance information will be lost; therefore, some similar points will be misclassified, which will cause the points that are far from each other in the high-dimensional space to gather together in the low-dimensional space after mapping. □

Given a high-dimensional data point $X = \{x_1, ..., x_n\}$ and a low-dimensional data point $Y = \{y_1, ..., y_n\}$. Using the nearest neighbor algorithm, the set of $k$ nearest neighbors of each $x_i$ is obtained $\{x_{i_1}, ..., x_{i_k}\}$. Where $x$ is a high-dimensional space, $x_i \in x$, $x_i$ denotes the $i$-th data in the high-dimensional space; then, using an exponential probability distribution, the high-dimensional topology can be expressed as follows.

$$p_{i|j} = \exp\left(\frac{-\max\left(0, d\left(x_i, x_{i_j}\right) - \rho_i\right)}{\sigma_i}\right) \tag{1}$$

where $\rho_i$ denotes the distance from point $x_i$ to the first nearest neighbor data point, and $\sigma_i$ denotes the diameter from point $x_i$ to the first nearest neighbor data point. Additionally, note that this is not a symmetric function, so this function should be symmetrized with high-dimensional probability to avoid overcrowding of cluster representations, so that different clusters can be represented in the overlapping regions.

$$p_{ij} = p_{i|j} + p_{j|i} - p_{i|j}p_{j|i} \tag{2}$$

High-dimensional probabilistic symmetry is necessary because after UMAP combines the points with local metric changes, it may appear that the weights of the graph between node $a$ and node $b$ are not equal to the weights between node $b$ and node $a$, where $p_{i|j}$ denotes the weight of the $i$-th point to the $j$-th point distance, and $p_{j|i}$ denotes the weight of the $j$-th point to the $i$-th point distance.

After establishing the topology in the high-dimensional spatial distribution, it is correspondingly necessary to establish the probability distribution in the low-dimensional space as well.

$$q_{ij} = \left(1 + a\left(y_i - y_j\right)^{2b}\right)^{-1} \tag{3}$$

The curve cluster $a(y_i - y_j)^{2b}$ is used in formula (3) to model the low-dimensional distance probabilities, not exactly t-distributed, where the default hyperparameters $a \approx 1.93$, $b \approx 0.79$ [5].

The UMAP algorithm expects data points of the same kind to be as close together as possible in the low-dimensional space after dimensionality reduction, while data points of different kinds are as far away from each other as possible. Therefore, the following function is introduced.

$$\text{Attractive} = p_{i|j}(X)\log\left(\frac{p_{i|j}(X)}{q_{i|j}(Y)}\right) \tag{4}$$

$$\text{Repulsive} = (1 - p_{i|j}(X))\log\left(\frac{1 - p_{i|j}(X)}{1 - q_{i|j}(Y)}\right) \tag{5}$$

In formulae (4) and (5), $p_{i|j}(X)$ is the weight of data points in the high-dimensional distribution, and $q_{i|j}(Y)$ is the weight of data points in the low-dimensional distribution. The algorithm first applies a gravitational force to data points of the same kind in the dataset and a repulsive force to data points of different kinds. Thus, it achieves the effect that data points of the same kind are clustered together, while data points of different kinds are kept away.

Given the above description, the specific description of the UMAP algorithm is shown in Algorithm 1.

---

**Algorithm 1.** UMAP algorithm

**function** UMAP ( $X$ , $n$ , $d$ , min-dist,n-epochs)

**for all** $x \in X$ **do**

fs-set [ $x$ ] $\leftarrow$ LocalFuzzySimplicialSet ( $X$ , $x$ , $n$ )

top-rep $\leftarrow \bigcup_{x \in X}$ fs-set [ $x$ ]

$Y \leftarrow$ SpectralEmbedding(top-rep, $d$ )

$Y \leftarrow$ OptimizeEmbedding(top-rep, $Y$ ,min-dist,n-epochs)

**return** $Y$

---

*3.2. Random Forest Algorithm Based on Parameter Optimization*

The random forest algorithm is a large-scale, high-dimensional data learning classifier integrated with multiple independent decision tree classifiers, each of which is obtained based on Bootstrap sampling. Then, multiple decision trees are combined to derive the final classification result by voting. The random forest algorithm is chosen to classify network traffic data because it can improve the training speed of large samples by high parallelization when dealing with the data volume of network security connection data. The process of constructing the random forest algorithm is as follows:

1.  Construct multiple sub-datasets. Form an intermediate dataset by selecting k samples through sampling with put-back from a dataset, including k samples, and then randomly select a few features among all features of this intermediate dataset as the final dataset;

2.  Build a sub-decision tree based on the sub-dataset. Assume that a sub-dataset has M features. When each node of the decision tree needs to split, randomly select m features from these features (m < M), then select one of the m features as the splitting attribute of the node. Keep repeating this step until it can no longer be classified. The principle of judgement is that the attribute selected by a node next time is the attribute used in the last split;

3.  Follow the above steps to construct a large number of sub-decision trees, which form a random forest;

4.  Input the dataset into different sub-decision trees. Different judgments will be obtained. The most judgmental result is the classification result obtained by the random forest.

Given the above description, the specific description of random forest algorithm is shown in Algorithm 2.

**Algorithm 2.** Random Forest algorithm

**For** $a = 1$ to $A$:

    (a) A boostrap sample $Y$ of size $X^*$ is randomly selected from the training set.

    (b) Grow a random-forest tree $T_a$ to the boostrapped data, repeating the following steps recursively for each terminal node of the tree, guiding to the minimum node size $x_{\min}$.

        i. Select $n$ variables at random from the $q$ variables.

        ii. Pick the best variable among the $n$.

        iii. Split the node into two child nodes.

Output the ensemble of trees $\{T_a\}_1^A$

To make a prediction at a new point $h$:

Classification: Let $\widehat{C}_a(h)$ be the class prediction of the $a$ th random-forest tree. Then,

$\widehat{C}_{\mathrm{rf}}^A(h) = majority \ vote \ \{\widehat{C}_a(h)\}_1^A$.

To further improve the effectiveness of the random forest algorithm for detecting malicious traffic in complex network traffic, this paper improves the random forest algorithm based on parameter optimization using base evaluators (n_estimators) and the maximum number of leaf nodes (max_leaf_nodes). Hyperparameters are very important for machine-learning algorithms because they directly control the behavior of the training algorithm and have an essential impact on the performance of machine-learning models.

*3.3. UMAP-RF Hybrid Algorithm*

This paper proposes a hybrid algorithm UMAP-RF, reduces the dimensionality of UNWS-NB15 dataset by the UMAP algorithm and thus improves the speed and accuracy of network attack traffic detection. The random forest algorithm is improved based on parameter optimization, and the dimensionality reduction data are bifurcated with the improved random forest algorithm. The abnormal data distinguished after bifurcation are classified to complete the network attack traffic detection task.

The UMAP algorithm is chosen as the dimensionality reduction method for the dataset in this paper mainly because it can preserve the data structure after the classification algorithm as much as possible. UMAP algorithm focuses more on the visualization of the classified data structure, and the UMAP visualization graph can intuitively represent the relationship between individual data features in the two-dimensional plane. For high-dimensional network-traffic-type datasets, this dimensionality reduction method is very effective for visualizing binary and classifying network attack traffic detection.

Improving the random forest algorithm based on parameter optimization aims to prevent overfitting when classifying more features in the UNSW-NB15 dataset and further enhance the random forest algorithm classification performance. The optimized random forest algorithm can effectively distinguish between normal and abnormal data and classify abnormal data. The accuracy obtained after classification is much higher compared to traditional machine-learning algorithms. The Figure 1 and Table 1 are training steps of the UMAP-RF hybrid algorithm and UMAP-RF hybrid algorithm flow chart.

1. Divide the UNSW-NB15 dataset into 70% training set, 10% validation set (Set hyperparameters) and 20% test set;

2. Convert the non-numerical features to numerical values and remove non-numerical features, such as proto, service and state, from the dataset to obtain dataset $\delta$;

3. The processed dataset $\delta$ is dimensionally reduced by the UMAP algorithm to obtain dataset $\theta$;

4. Optimize the parameters of the random forest algorithm with the number of base evaluators and the maximum number of leaf nodes. n_estimators is set to 1000, and max_leaf_nodes is set to 10 by manual parameter tuning;

5. Import the dataset $\theta$ obtained after UMAP dimensionality reduction into the improved random forest algorithm for training. Classify normal data $\alpha$ and abnormal data $\beta$, and complete malicious traffic detection;

6. Import the detected abnormal data $\beta$ into the improved random forest algorithm to classify the abnormal data $\beta$, and obtain each specific type of network attack traffic.

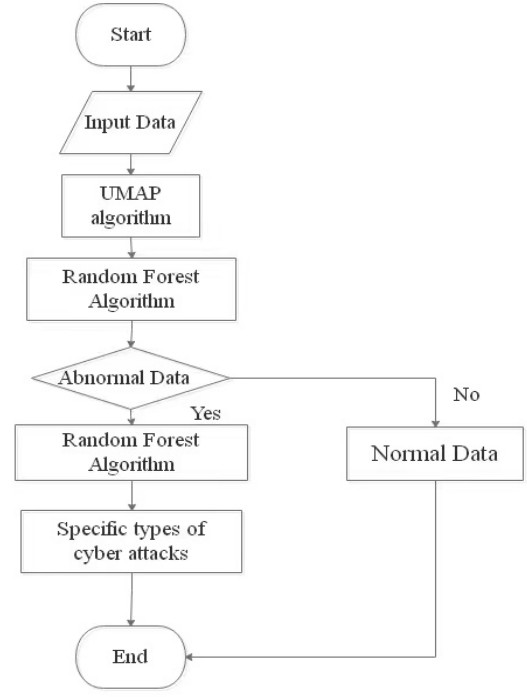

**Figure 1.** UMAP hybrid algorithm.

**Table 1.** List of features in the UNSW-NB15 dataset.

| No. | Feature | No. | Feature | No. | Feature | No. | Feature | No. | Feature |
|-----|---------|-----|---------|-----|---------|-----|---------|-----|---------|
| 1 | srcip | 11 | dttl | 21 | stcpb | 31 | sintpkt | 41 | ct_srv_src |
| 2 | sport | 12 | sloss | 22 | dtcpb | 32 | dintpkt | 42 | ct_srv_dst |
| 3 | dstip | 13 | dloss | 23 | smeansz | 33 | tcprtt | 43 | ct_dst_ltm |
| 4 | dsport | 14 | service | 24 | dmeansz | 34 | synack | 44 | ct_src_ltm |
| 5 | proto | 15 | sload | 25 | trans_depth | 35 | ackdat | 45 | ct_src_dport_ltm |
| 6 | state | 16 | dload | 26 | res_bdy_len | 36 | is_sm_ips_ports | 46 | ct_dst_sport_ltm |
| 7 | dur | 17 | spkts | 27 | sjit | 37 | ct_state_ttl | 47 | ct_dst_src_ltm |
| 8 | sbytes | 18 | dpkts | 28 | djit | 38 | ct_flw_http_mthd | 48 | attack_cat |
| 9 | dbytes | 19 | swin | 29 | stime | 39 | is_ftp_login | 49 | label |
| 10 | sttl | 20 | dwin | 30 | ltime | 40 | ct_ftp_cmd | | |

## 4. Experiment and Analysis

### 4.1. UNSW-NB15 Dataset and Experimental Environment

The UNSW-NB15 dataset, created by the Australian Centre for Cyber Security (ACCS) Network Scope Lab, is the closest simulation of the current network traffic environment, covering one regular class of data and nine attack classes of data. Denial of Service (Dos), Exploits (Attack), Generic (General), Reconnaissance, Shellcode and Worms. As shown in Table 1, features 1 to 5 are stream features collected from the dataset, 6 to 18 are based features, 19 to 26 are content features, 27 to 35 are temporal features, 36 to 40 are generic features, 41 to 47 are general features and 48 to 49 are label features. In this paper, we use the training set of UNSW-NB15 to train the model and use its test set to evaluate the model performance.

The network attack traffic detection model experiments and comparison experiments presented in this paper were conducted on a 64-bit Windows Intel(R) Core (TM) i7-11700K CPU (3.60 GHz) with 32 GB RAM and a Python-based Nvidia GeForce GTX 3070 GPU (8 GB), using Python's TensorFlow library to write the UMAP-RF model for this paper.

### 4.2. Experimental Results of Dimensionality Reduction Algorithm

The pre-processed data are imported into the PCA algorithm [25], T-SNE algorithm [26] and UMAP algorithm, all of which are commonly used in machine learning to process the dimensionality reduction in high-dimensional datasets. The visualization plots and running times of the three-dimensionality reduction algorithms are derived by experimental simulation, respectively.

Figure 2 show the data structure visualization after being processed by the dimensionality reduction algorithm PCA, T-SNE and UMAP, respectively. The purple color at the top of the color gradient bar on the right side of the figure to the eighth orange color from the top down are abnormal attack data. Different color blocks represent different types of attacks, and the rosy color at the bottom is normal data. The more concentrated points of the same color and the more separate issues of different colors indicate the apparent effect of data clustering of each feature, thus reflecting the impact of dimensionality reduction algorithm clustering. The comparison from the figure shows that the UMAP algorithm has the best effect of dimensionality reduction on the dataset and effectively spreads out the data of different attack types.

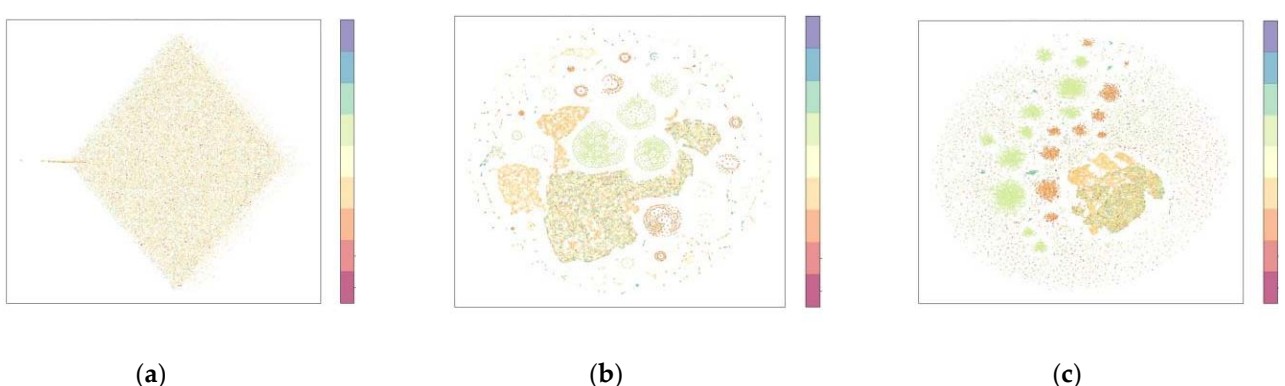

(**a**)　　　　　　　　　　　(**b**)　　　　　　　　　　　(**c**)

**Figure 2.** Visualization of data structure after dimensionality reduction by three different algorithms. (**a**) PCA algorithm. (**b**) T-SNE algorithm. (**c**) UMAP algorithm.

As shown in Figure 2, although the principal component analysis (PCA) method projects the high-dimensional data onto a two-dimensional plane after dimensionality reduction, all features are crowded together without bothering to distinguish normal data from abnormal data. Therefore, the principal component analysis (PCA) method is ineffective in dimensionality reduction for network attack traffic. As shown in Figure 3, T-SNE can further distinguish between normal and abnormal data and preserve the data structure

after dimensionality reduction. However, T-SNE is not applicable to large samples and cannot achieve the preservation of the global structure. Compared with T-SNE, the UMAP algorithm can better reflect the high-dimensional data structure with better continuity. As shown in Figure 4, the distinction between normal and abnormal data is more obvious in the two-dimensional plane, while the global structure is better preserved after the dimensionality reduction.

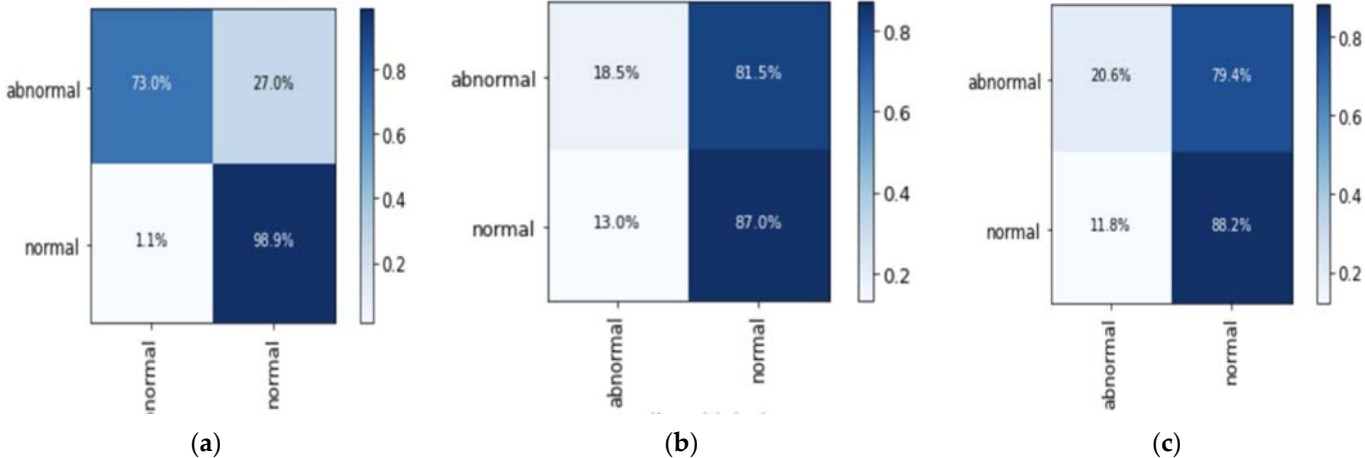

**Figure 3.** The confusion matrix by three different algorithms. (**a**) UMAP-RF algorithm. (**b**) LOF algorithm. (**c**) Isolation Forest algorithm.

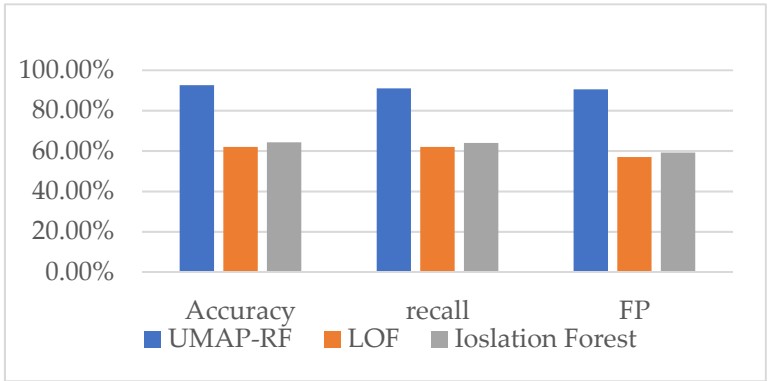

**Figure 4.** Histogram comparing the results of the binary classification experiment.

This paper also compares the running time of three dimensionality reduction algorithms, as shown in Table 2. Principal component analysis (PCA) has the shortest running time and is much shorter than the other two methods, but it cannot show the relationship between the data after dimensionality reduction. The network traffic dataset needs to retain the structure of the reduced data as much as possible after dimensionality reduction, so principal component analysis (PCA) cannot be applied to the study in this paper. Compared with the running time of 695.31s of T-SNE algorithm, the running time of UMAP algorithm is only 93.82s, which is much smaller than that of T-SNE algorithm, and UMAP algorithm can better retain the data structure after dimensionality reduction.

**Table 2.** Running time of three-dimensionality reduction algorithms.

| Algorithm | Running Time |
|-----------|--------------|
| PCA | 3.59 s |
| T-SNE | 695.31 s |
| UMAP | 93.82 s |

In this paper, the UMAP algorithm is introduced for the first time to reduce the dimensionality of network traffic data, which can not only preserve the data structure to the maximum extent for visualization and research but can be processed very fast when facing massive network traffic data.

### 4.3. Experimental Evaluation Metrics

In this paper, the classification experiments evaluate the performance of the intrusion detection model using four metrics, such as area under curve (AUC), accuracy (accuracy, ACC), recall (recall) and F1 score (F1 Score).

The confusion matrix is a situation analysis table for summarizing the prediction results of classification models in machine learning. As shown in Table 3, TP is the number of correctly predicted positive cases, which is the number of connection records correctly classified as attack class in network attack security detection. FN is the number of incorrectly predicted positive cases, which is the number of connection records incorrectly classified as normal class in the scenario of this paper. FP is the number of incorrectly predicted negative cases, which is the number of connection records incorrectly classified as attack class in the scenario of this paper. TN is the number of correctly predicted negative cases, which in the scenario of this paper is the number of connection records correctly classified as normal class.

**Table 3.** Confusion matrix.

| | | Projections | |
|---|---|---|---|
| | | Attack | Normal |
| **Actual** | Attack | TP | FN |
| | Normal | FP | TN |

The AUC metric and the ROC curve are introduced in binary classification because they provide a more comprehensive assessment of a classifier's, and in particular, the classifier's effectiveness in binary classification, where area under curve (AUC) is defined as the area under the ROC curve.

Accuracy, Recall and F1 Score are defined as formulas (6)–(8), respectively.

$$\text{Accuracy} = \frac{TP + TN}{TP + TN + FP + FN} \tag{6}$$

$$\text{Recall} = \frac{TP}{TP + FP} \tag{7}$$

$$\text{F1 Score} = \frac{2 \times \text{Recall} \times \text{Precision}}{\text{Recall} + \text{Precision}} \tag{8}$$

### 4.4. Random Forest Classifier Parameter Settings

For the random forest algorithm classifier, the larger the number of base evaluators (n_estimators) in general, the better the classification effect of the algorithm, but all the computation and memory required are also larger, which will lead to an increase in training time. Due to many feature components in the UNSW-NB15 dataset, it is easy to overfit

in the process of classification. Overfitting can be prevented by limiting the maximum number of leaf nodes (max_leaf_nodes).

In this paper, we perform parameter tuning on the validation set by manually tuning the parameters and finally select the highest scoring set as the final hyperparameters of the random forest algorithm. The scores of each hyperparameter group obtained by manual tuning are shown in Table 4.

**Table 4.** Scores of each hyperparameter group obtained by manual tuning.

| No. | Score | (n_estimators, max_leaf_nodes) |
|---|---|---|
| 1 | 83.7 | (800, 30) |
| 2 | 86.2 | (1600, 40) |
| 3 | 91.6 | (1000, 10) |
| 4 | 77.5 | (500, 70) |
| 5 | 81.1 | (2000, 20) |
| 6 | 85.4 | (200, 10) |

As shown in Table 4, the third set of hyperparameters scored the highest, so the n_estimators best parameter is 1000, and the max_leaf_nodes best parameter is 10. The random forest algorithm will build the optimal decision tree within the optimal set of parameters set in this paper.

### 4.5. Analysis of Binary Classification Experimental Results

The confusion matrix derived from the hybrid algorithm UMAP-RF, isolated forest (IF) [27] and local outlier factor (LOF) [28] on a test set is shown in Figure 3. Based on the parameters in this result, several evaluation indices were experimentally calculated in anticipation of obtaining a comprehensive performance report of the algorithm.

In Figure 3, the upper left part of the confusion matrix shows the number of connection records correctly classified as attack class, and the upper right part shows the number of connection records of normal class. The lower left part shows the number of connection records incorrectly classified as attack class. The lower right part shows the number of connection records correctly classified as normal class. The accuracy rate of each algorithm can be calculated from formula (6), the regression rate of each algorithm can be calculated from formula (7), and the F1 score of each algorithm can be calculated from formula (8).

Based on the confusion matrix plots of the above three algorithms, the AUC, accuracy, recall and F1 scores of the hybrid algorithm UMAP-RF hybrid, isolated forest algorithm and LOF algorithm are derived and compared, and the comparison of the experimental results is shown in Figure 4.

As shown in Figure 4, the accuracy, recall and F1 score of the hybrid algorithm UMAP-RF based on binary classification experiments are significantly improved compared with other machine-learning algorithms, with an accuracy of 92.6%, a recall of 91% and an F1 score of 90.5%, demonstrating that the hybrid algorithm UMAP-RF has a better detection effect for distinguishing normal data from abnormal data.

The ROC curves for the hybrid algorithm UMAP-RF, isolated forest algorithm and LOF algorithm are shown in Figure 5. The AUC value is the area under the ROC curve. The calculation shows that the AUC value for the hybrid algorithm UMAP-RF is 85%, the isolated forest algorithm is 54.4%, and the LOF algorithm is 52.7%. Based on the AUC values, it can be seen that the hybrid algorithm UMAP-RF performs significantly better compared to the isolated forest algorithm and the LOF algorithm.

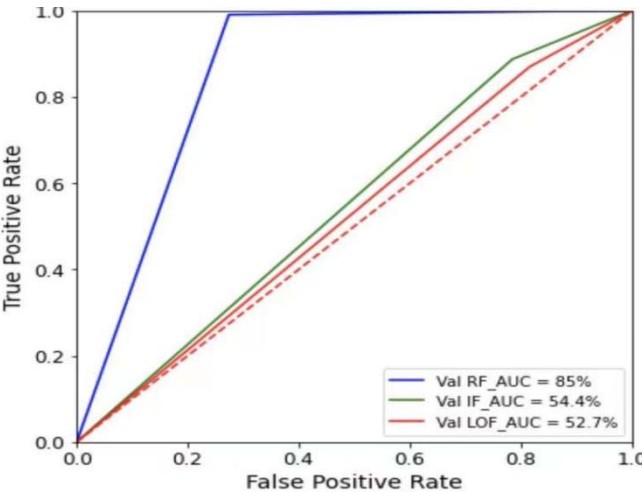

**Figure 5.** ROC curve of binary classification experiments.

To demonstrate the effect of the hybrid algorithm UMAP-RF classifier more visually, a visualization study of the binary classification-based hybrid algorithm UMAP-RF classifier was conducted. As shown in Figure 6, the purple part represents normal network traffic data points, and the rosy part represents abnormal attack traffic data points. The purple data points are clustered together and separated from the rosy data points in the figure, which maximally restores the data structure after binary classification and demonstrates a better clustering effect for visualization and research.

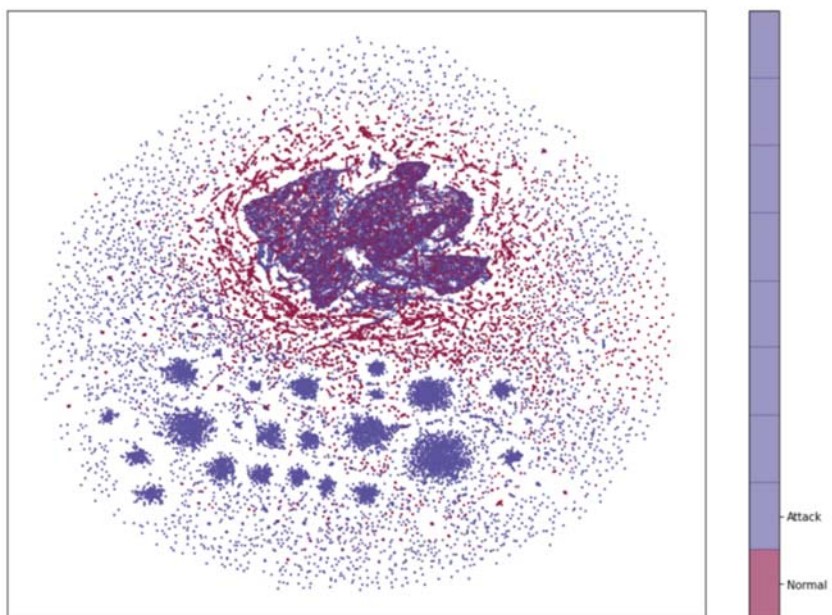

**Figure 6.** Visualization of binary classification effect based on hybrid algorithm UMAP-RF.

To demonstrate the applicability of the hybrid algorithm UMAP-RF in network attack traffic detection, the normal and abnormal data in the dataset of KDDCUP99 and NSLKDD were classified by the hybrid algorithm UMAP-RF, LOF algorithm and isolated forest algorithm in the experimental part. The accuracy, recall and F1 score obtained after classification were also compared, and the experimental results of different algorithms are shown in Figures 7 and 8.

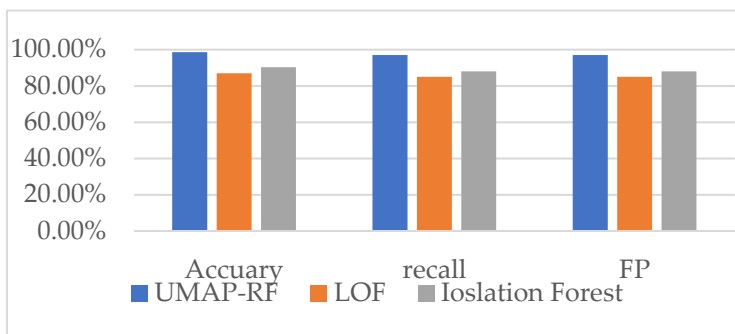

**Figure 7.** Comparison histogram of experimental results of different algorithms for KDDCUP dataset.

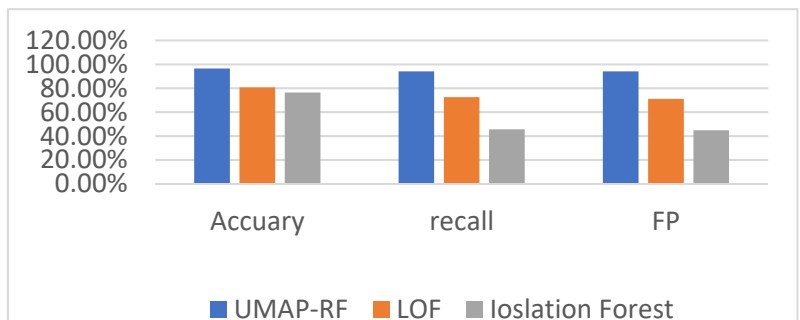

**Figure 8.** Comparison histogram of experimental results of different algorithms for NSL-KDD dataset.

As can be seen from Figures 7 and 8, the accuracy, recall and F1 score of the hybrid algorithm UMAP-RF after classifying normal and abnormal data in the KDDCUP99 dataset and USL-KDD dataset are significantly improved compared with LOF and isolated forest algorithms. The hybrid algorithm UMAP-RF still has good results in distinguishing normal and abnormal data in the KDDCUP99 dataset, proving that the hybrid algorithm UMAP-RF has good applicability in network attack traffic detection.

The experimental comparison and analysis results above show that the hybrid algorithm UMAP-RF is feasible in the study of the intelligent network attack traffic detection model. To further demonstrate the performance of the hybrid algorithm UMAP-RF, the differentiated anomalous data are classified again with specific network attack types to prove the feasibility of the hybrid algorithm UMAP-RF in multi-classification of network attack traffic.

*4.6. Analysis of Multi-Classification Experimental Results*

To evaluate the hybrid algorithm UMAP-RF in the multi-classification task of network attack traffic detection, the results of the control experiments are used for comparative analysis, and the algorithms used in the control experiments are the K-Means clustering algorithm [29], mini batch K-Means algorithm [30] and LSTM [31] neural network algorithm. The confusion matrix derived from the hybrid algorithm UMAP-RF and the algorithm used in the control experiment is shown in Figure 8. Based on the parameters in this result, several evaluation metrics were calculated for the experiment, and since the AUC metric does not apply to the multi-classification task, only the accuracy, recall and F1 score were used as evaluation metrics for the experiment in the expectation of obtaining a comprehensive performance report of the algorithm.

Figure 9 shows the multi-classification confusion matrix for the K-Means algorithm, mini batch K-Means algorithm, LSTM algorithm and hybrid algorithm UMAP-RF. The multi-classification confusion matrix is calculated in the same way as the binary confusion matrix. The accuracy rate, regression rate and F1 score of each attack type are calculated

from formula (6)–(8). The multi-classification metrics of each algorithm can be obtained by averaging the metrics of all attack types.

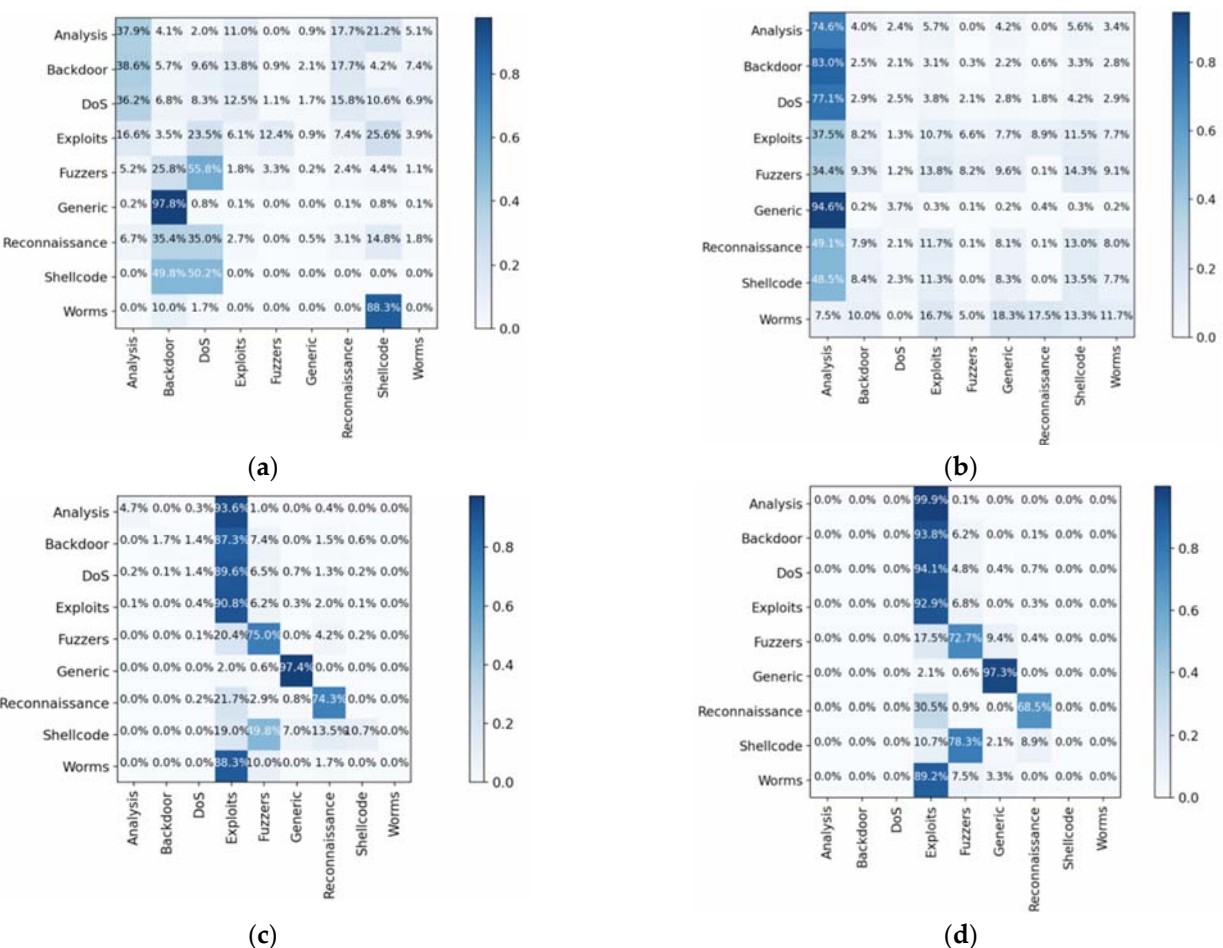

(**a**)

(**b**)

(**c**)

(**d**)

**Figure 9.** The confusion matrix by four different algorithms. (**a**) K-Means algorithm. (**b**) Mini Batch K-Means algorithm. (**c**) LSTM algorithm. (**d**) UMAP-RF algorithm.

Based on the confusion matrix plots of the above four algorithms, the accuracy, recall and F1 scores of the hybrid algorithms UMAP-R and K-Means algorithm, mini batch K-Means algorithm and LSTM neural network algorithm are derived and compared, and the experimental results are shown in Figure 10.

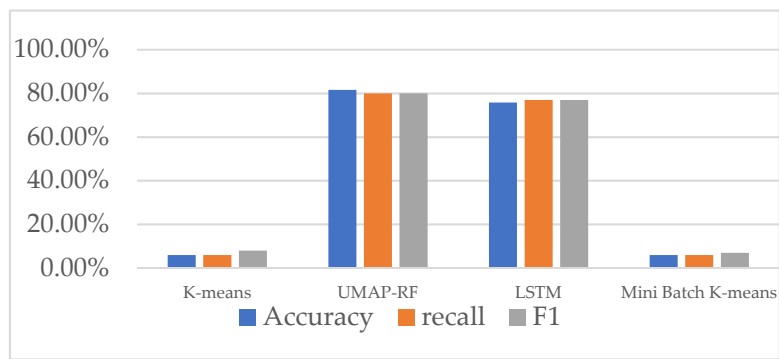

**Figure 10.** Histogram comparing the results of the multi-classification experiment.

As can be seen from Figure 10, the accuracy, recall and F1 score of the hybrid algorithm UMAP-RF based on abnormal data classification are significantly improved compared with other machine-learning algorithms, with an accuracy of 81.6%, a recall of 80%

and an F1 score of 80%, respectively, demonstrating that the hybrid algorithm UMAP-RF still has a better detection effect for the differentiation of anomalous data.

To demonstrate the effect of the hybrid algorithm UMAP-RF classifier more visually, the hybrid algorithm classifier based on multi-classification of abnormal data was visualized by the UMAP algorithm. As shown in Figure 11, the color bar on the right of the figure denotes different attack types with different colors. The data points of the same attack type are clustered together in the figure, while data points of different attack types are separated from each other, which maximally restores the data structure after multi-classification and demonstrates a better clustering effect to make it convenient for visualization research.

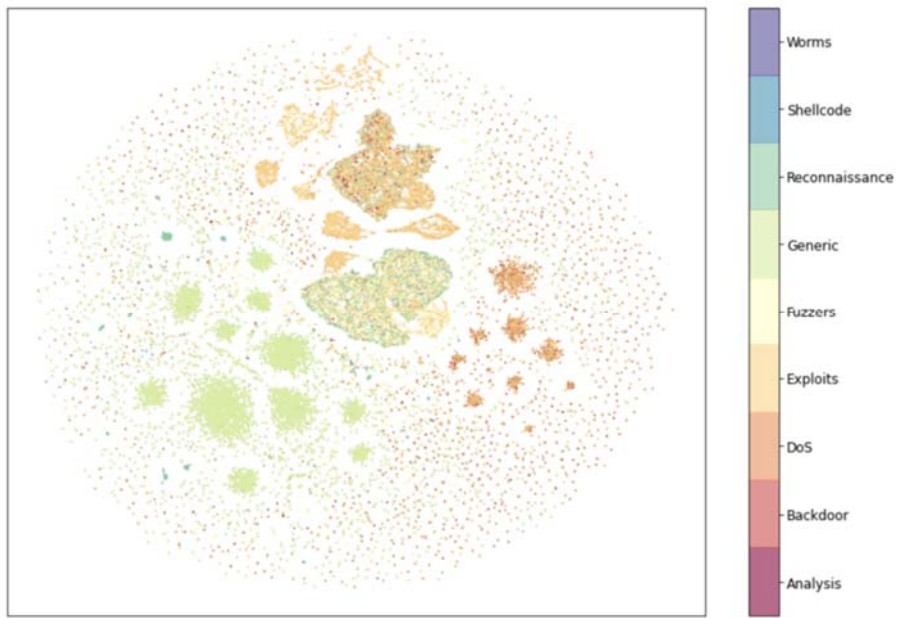

**Figure 11.** Visualization of multi-classification effect based on hybrid algorithm UMAP-RF.

The experimental comparison and analysis results above show that the hybrid algorithm UMAP-RF is also effective for multi-classification of abnormal data.

*4.7. Comparison of the Running Time of UMAP-RF Hybrid Algorithm and Other Algorithms*

Time is a critical factor in preventing and reducing the effects of the attack. The network security connection data after dimensionality reduction by the UMAP algorithm highlights key feature information and effectively clusters data with different features, which significantly reduces the time required for subsequent detection of abnormal data and directly improves the efficiency of network traffic attack detection. In the binary classification experimental section, the running time of the hybrid algorithm UMAP-RF (including the running time of UMAP dimensionality reduction) is compared with other machine-learning algorithms, as shown in Table 5.

**Table 5.** Running time comparison of different algorithms.

| Algorithm | Running Time |
|---|---|
| RF | 379.23 s |
| LOF | 1457.09 s |
| Isolation Forest | 1027.12 s |
| UMAP-RF | 243.39 s |

As shown in Table 5, the UMAP algorithm reduces the dimensionality and then classifies by the random forest algorithm compared to the direct random forest algorithm in terms of running time by one-third, proving that the UMAP dimensionality reduction algorithm can significantly reduce the subsequent time for detecting abnormal data. The hybrid algorithm UMAP-RF is not only more accurate than the other two algorithms in detecting abnormal data but also requires the lowest running time. By comparing the running time and accuracy of different algorithms in dichotomous experiments, it is demonstrated that the hybrid algorithm UMAP-RF can effectively detect abnormal data in a shorter time and outperform the other algorithms in terms of comprehensive performance.

### 4.8. Time Complexity of the Hybrid Algorithm UMAP-RF

The main factors that determine the time complexity of the dimensionality reduction algorithm UMAP include：the number of data points in the high-dimensional space is $n$, the original dimension of the high dimension is $D$, and the target dimension of the low dimension is $d$. The time complexity of the computational process of mapping the data points in the high-dimensional space to the low-dimensional space is $O（nD）$; therefore, the time complexity of the UMAP algorithm is $O（nD）$.

The main factors that determine the time complexity of the random forest algorithm include: the size of sample is $N$, the number of features is $M$, and the depth of the tree is $F$. When the cart grows, all the values within the feature are taken as split candidates, and an evaluation index is calculated for them, so the time complexity of each layer is $O（NM）$, and the time complexity of the tree at layer $F$ is $O（NMF）$; therefore, the time complexity of the random forest algorithm is $O（NMF）$.

As shown above, the time complexities of the UMAP algorithm and the random forest algorithm are of the same order of magnitude, so the time complexity of the hybrid algorithm UMAP-RF is $O（nD + NMF）$.

### 4.9. Comparison of the Detection Effect of UMAP-RF Hybrid Algorithm and Other Algorithms

Because the algorithms used in the control experiments are traditional machine-learning methods, the reference is insufficient. This paper also collects the latest results on network attack traffic detection at home and abroad. To ensure the fairness of the experimental comparison, the experimental results compared are used in the UNSW-NB15 dataset. The comparison between the hybrid algorithm UMAP-RF and the detection effect of classification algorithms used by Guoyan Huang's team [23], Fengjie Hu [26], Meftah [27], Kasongo [28], Cao Bo [29] and Alzaqebah [30] is shown in Table 6 below.

**Table 6.** Comparison of the detection effect of UMAP-RF hybrid algorithm and other algorithms.

| Literature | Algorithm | ACC/% |
|---|---|---|
| Guoyan Huang [23] | (LR-RFE) + DT | 88.27% |
| Fengjie Hu [26] | light GBM | 85.78% |
| Meftah [27] | SVM | 82.00% |
| Kasongo [28] | XGBOOST+DT | 90.85% |
| Cao Bo [29] | CNN-GRU | 86.25% |
| Alzaqebah [30] | GWO-ELM | 81.00% |
| Proposed in this paper | UMAP-RF | 92.60% |

Table 6 demonstrates that the hybrid algorithm UMAP-RF has a significant improvement in accuracy compared to other algorithms, with optimal detection results, further proving that the hybrid algorithm UMAP-RF is practical and feasible in network attack traffic detection.

## 5. Conclusions

This paper applies machine-learning algorithms to network attack traffic detection and proposes a hybrid algorithm UMAP-RF for binary and multi-classification network attack detection tasks. The dataset used in this paper is the public dataset UNSW-NB15. This paper uses the UMAP dimensionality reduction algorithm to reduce the dimensionality of the high-dimensional dataset, which significantly accelerates the detection speed and accuracy of the network attack traffic. Additionally, the random forest algorithm is improved based on parameter optimization, using the base evaluator parameter and the maximum leaf node parameter to prevent overfitting and improve the classification performance of the random algorithm. At the same time, the hybrid algorithm UMAP-RF has significant advantage in accuracy compared with the control experimental results and the research results in recent years at home and abroad in the binary and multi-classification experiments.

The research in this paper also has some shortcomings because there are no conditions to build a large data network center, so the solution proposed in this paper cannot be experimentally analyzed in a practical application environment. We will make this part of my work the focus of future research and will seek to cooperate with some large domestic data network centers to further improve this research.

**Author Contributions:** Resources, Y.W. and Z.H.; Methodology, X.D. and C.C.; Validation, C.C.; Writing—Original Draft Preparation, X.D. and C.C.; Writing—Review and Editing, X.D. and C.C.; Supervision, Z.H. and Y.W. All authors have read and agreed to the published version of the manuscript.

**Funding:** This work was supported by National Natural Science Foundation of China (61701170), Special project for key R&D and promotion of Henan Province (222102210052, 222102210007, 222102210062 and 222102210272).

**Data Availability Statement:** The datasets are available in a publicly accessible repository at https://research.unsw.edu.au/projects/unsw-nb15-dataset (accessed on 13 August 2021). Additionally, the perfect code data for the proposed method in this paper can be obtained by contacting the authors of this study.

**Conflicts of Interest:** The authors declare no conflicts of interest.

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
