# Peer review of "Research on Network Attack Traffic Detection HybridAlgorithm Based on UMAP-RF"

_algorithms, doi:10.3390/a15070238_

Round 1

Reviewer 1 Report

This paper is very well written, I have only one concern and that is about figure-1. The text size is very small.

Reviewer 2 Report

1. The paper structure should be described at the end of the introduction section.

2. The introduction should indicate the research gaps and research goals. What does it add to the subject area compared with other published material?

3. Do you consider the topic original or relevant in the field, and if so, why?

4. What specific improvements could the authors consider regarding the methodology?

5. The implementation environment is not clear.

6.   The solution is described but there should be more examples.

7.   The description of the proposed solution should be more formal.

8.   The authors should add proof(s) of the properties, theorem or lemmas contained in the paper.

9.   The algorithm(s) should be described clearly using pseudocode.

10.     It is necessary to discuss the complexity of the proposed solution.

11.     Some experiment(s) should be added to show that the proposed solution can be used in real applications.

Reviewer 3 Report

Time is a critical factor in preventing and reducing the effects of the attack so this comparison regarding the running time should be made in parallel with the accuracy to decide that the proposed algorithm is indeed better than other algorithms.

Authors should specify the meaning of all acronyms first time they use it.

Round 2

Reviewer 2 Report

The authors have fixed previous concerns.